# High Nitrate and Phosphate Ions Reduction in Modified Low Salinity Fresh Water through Microalgae Cultivation

**Ahmad Rozaimee Mustaffa [1,2], Ku Halim Ku Hamid [2], Mohibah Musa [2], Juferi Idris [2,3,*] and Roslina Ramli [4]**

[1] Faculty of Chemical Engineering, Universiti Teknologi MARA (UiTM) Terengganu, Bukit Besi 23300, Dungun, Terengganu, Malaysia; rozaimee190@tganu.uitm.edu.my

[2] Faculty of Chemical Engineering, Universiti Teknologi MARA (UiTM), Shah Alam 40450, Selangor, Malaysia; kuhalim@salam.uitm.edu.my (K.H.K.H.); mohibah@salam.uitm.edu.my (M.M.)

[3] Faculty of Chemical Engineering, Universiti Teknologi MARA (UiTM), Sarawak Branch, Samarahan Campus, Kota Samarahan 94300, Sarawak, Malaysia

[4] Faculty of Computer Science and Mathematics, Universiti Teknologi MARA (UiTM) Terengganu, Dungun 23000, Terengganu, Malaysia; roslina_ramli@tganu.uitm.edu.my

* Correspondence: juferi@sarawak.uitm.edu.my; Tel.: +6082-677835; Fax: +6082-677300

**Abstract:** The treatment of nitrate and phosphate ions in fresh water such as streams, rivers, lakes, reservoirs, and wetlands through biological treatment has been getting more crucial and popular in recent years. This paper reports the reduction of nitrate and phosphate ions in modified low salinity fresh water via the cultivation of a microalgae strain, e.g., *Tetraspora* sp. and *Spirogyra* sp. The low salinity fresh water (9054 to 9992 ppm) was modified with the addition of organic fertiliser (VermiCompost Tea) and used as the cultivation medium to grow microalgae. The microalgae strains were grown under controlled conditions in an indoor laboratory for 14 days. The initial concentrations of nitrate and phosphate ions in the modified fresh water sample were 1.17 mg/L and 0.10 mg/L, respectively. The reduction of nitrate and phosphate ions associated with the effect of cultivation of *Tetraspora* sp. and *Spirogyra* sp. in ambient air (0.03% of $CO_2$) and 15% of $CO_2$ was investigated. In ambient air, the cultivation of *Tetraspora* sp. and *Spirogyra* sp. greatly reduced the nitrate ions concentration from $5.96 \pm 0.28$ to $0.37 \pm 0.05$ mg/L and from $2.35 \pm 0.19$ to $0.59 \pm 0.08$ mg/L, respectively. A 100% reduction of phosphate ions was observed in the cultivation of *Tetraspora* sp. and *Spirogyra* sp. from $0.52 \pm 0.10$ mg/L in 13 days of and from $0.63 \pm 0.15$ mg/L in 6 days, respectively. Meanwhile, with the aeration of 15% of $CO_2$, after the 14 days cultivation of *Tetraspora* sp. and *Spirogyra* sp. reduced the nitrate ions concentration from $5.27 \pm 0.06$ to $1.80 \pm 0.20$ mg/L and from $4.73 \pm 0.12$ to $2.80 \pm 0.10$ mg/L, respectively. The excessive $CO_2$ in water consequently lowered the pH of water medium from 7.18 to 6.60 due to the formation of carbonic acid ($H_2CO_3$). It was feasible to couple the removal of nitrogen and phosphorus in Sungai Sura (4°42″28.2° N 103°26″12.1° E) while cultivating microalgae through biological treatment to produce biomass for biofuel production.

**Keywords:** phosphate ions; nitrate ions; carbon dioxide; fresh water; low salinity; *Tetraspora* sp.; *Spirogyra* sp.; biodiesel

## 1. Introduction

The increase and advanced living standards of the world's population causes a high level of water consumption every year. This phenomenon can lead to enhanced utilisation of water that also contributes to water pollution. Wastewater can originate from a combination of domestic, municipal,

surface runoff or storm water, agricultural and industrial sources. Wastewater generally contains organic masses like proteins, carbohydrates, lipids, volatile acids and inorganic content (macronutrients and micronutrients) containing nitrate ions, phosphate ions, sodium, calcium, potassium, magnesium, chlorine, sulfur, bicarbonate, ammonium salts, and heavy metals [1,2]. These excessive nutrients in surrounding water bodies will encourage eutrophication or algae blooms due to the anthropogenic waste production. Furthermore, the algae accumulation becomes a serious problem particularly at the end of the growing season when most of the algae dies in the aqua-system [3]. The algae decomposition in streams, lakes or ponds (aquatic systems) consequently reduces the amount of dissolved oxygen that sustains eutrophication, increases odour and unpleasant tastes, and kills aquatic life [4]. Most developing nations do not have enough water treatment facilities where the main sources of wastewater effluent produced by domestic, municipal, agricultural and industrial activities are primarily released into the environment without treatment.

Several species of microalgae are able to adapt and grow efficiently in wastewater environments through their capability to treat and uptake the abundant natural inorganic compounds such as nitrate, phosphate, magnesium, sodium, calcium, potassium and heavy metals in the wastewater [5,6]. In addition, domestic effluent containing high concentrations of phosphate and nitrate ions partially coming from residential biodegradable household waste such as detergent in drainage or streams [7]. The waste water effluents produced from intensive agricultural land also encourage the growth of aquatic plants such as microalgae in aquaculture systems [8,9]. This situation may effect water flowing systems and the related public authorities have to allocate expenses to remove the aquatic plant growth in the effluents or streams to ensure smooth water flow [10].

Most microalgae species can act as waste-to-wealth converters and recyclers, as they use and recycle nutrients and remain available instead of polluting the water. They can produce valuable raw materials such as biofuel (biodiesel, bioethanol and biogas), proteins, starch and pigments such as β-carotene, tocopherols and terpenes [11]. The applications of these materials are numerous, ranging from biodiesel and bioplastics to colorants and food to dietary supplement products such as C-phycocyanin, astaxanthin, and β-carotene, which is a source of vitamin A and can be used as a food colouring agent as well as cosmetic additives [12]. The maximum productivity of biomass can be achieved with 10% of $CO_2$ aeration that produces the highest algae populations. The increase in population biomass can be reached at 5% of $CO_2$ [13].

Microalgae can be applied as a bio-aqua plant that is capable of treating domestic wastewater such as oxidation ponds, lakes, and river water since they are adapted to grow in a wide range of environmental conditions. This species has also been deployed as an environmentally friendly wastewater treatment at a reasonable cost [14]. A main constraint for wastewater-based algae biofuel production system is discovering the most ideal microalgae strains which are able to grow in the wastewater environment with significant nutrient removal efficiency showing high biomass and lipid production [15]. Microalgae cells also have the ability to uptake nitrogen and phosphorus from water. These species can be efficiently used to eliminate or reduce a significant amount of nutrients because they need high amounts of nitrogen and phosphorus [15].

Studies on nitrate and phosphate ion removal through cultivating microalgae strains such as *Tetraspora* sp. and *Spirogyra* sp. obtained from Sungai Sura fresh water are still limited. The study of the effect of $CO_2$ on algal culture is also limited. Thus, the objective of this study is to determine the ability of common local microalgae strains (*Tetraspora* sp. and *Spirogyra* sp.) to reduce the concentration of nitrate and phosphate ion via cultivation in aquatic systems, focusing on the effect of aeration using ambient air (0.03% of $CO_2$) and an additional 15% of $CO_2$.

## 2. Materials and Methods

### 2.1. Strains and Culture Medium

In this study, the green microalgae (chlorophyta) strains, namely *Spirogyra* sp. and *Tetraspora* sp., were obtained from local Sungai Sura river water flows in Dungun, Terengganu, Malaysia (4°42″28.2° N 103°26″12.1° E) about 1 km outside (upper estuary) Universiti Teknologi MARA campus Sura Hujung in Dungun, Terengganu, Malaysia. The river water could be contaminated with waste water from small-medium agricultural farm activities and residential domestic water that eventually flows to the sea. Thus, Sungai Sura downstream water can be classified as being in the low salinity range of fresh water with a salinity range of 9054 to 9992 ppm. The microalgae strains were stored at 10–15 °C to ensure the cells can grow after transferring to the culture medium [16]. These species were selected for cultivation due to their dominance and being the most commonly found in Sungai Sura fresh water. The collected water was then used as a culture medium for the determination of water quality parameters.

### 2.2. Sampling Raw Water Method

The samples of fresh water were collected downstream of Sungai Sura. The sampling was conducted 15–30 cm below the water surface for water quality analysis. Two raw Sungai Sura water samples were collected using 500 mL wide-mouth plastic bottles and transferred into 1500 mL transparent plastic bottles. The samples were stored at 10–15 °C for preservation purposes. The parameters of water sample analysed were pH, nitrate ions, phosphate ions, total suspended solids and dissolved oxygen (DO).

### 2.3. Addition of Organic Fertiliser

The fresh water medium (3 L) was added with 5 mL of organic fertiliser. This process was implemented due to the low nitrate and phosphate ions present in the raw water. Then, 5 mL of water- based organic fertiliser; namely VermiCompost Tea Organic Liquid Fertilizer with 100% natural and organic liquid plant food was added into each cultivation flask. VermiCompost Tea was brewed using vermicompost and contained beneficial soil micro-organisms which can be applied directly in various agricultural and horticultural applications. This liquid fertiliser was supplied by Tam Por Joo, a local company located in Selangior, Malaysia. This fertiliser consisted of organic carbon (9.8–13.4%), nitrate (0.15–0.73%), phosphate (0.19–1.02%), potassium (0.15–0.73%), calcium (1.18–7.61%) and other nutrient elements [17]. Then 10 mL of water was collected to determine the nitrate and phosphate ion concentrations using a HACH UV-visible spectrophotometer (Loveland, CO, USA) model DR900 based on the 8039 cadmium reduction method for nitrate ions and the 8048 method for phosphate ions.

### 2.4. Ozone Treatment of Culture Medium Preparation

The fresh water from Sungai Sura was used as the growing medium for microalgae strains that are classified as low salinity water due to the rivers' intercept with sea water (Open University Course Team, 1999). The prepared medium was disinfected for 60 min in an ozoniser (Hirayama HV-50, Tokyo Japan) exposing the water to ozone (2 mg/L) to kill bacteria and filter out a wide range of contaminants. Then, the microalgae cells were transferred into a 500 mL conical flask, the culture medium consisted of low salinity fresh water and 5 mL of water-based organic fertiliser were added to each flask. All treatments and blank water were conducted in triplicate and the experiment was done in 14 days. The initial pH of medium was set up, monitored and recorded at a range of 6.60 to 7.18. This process can also be applied in the removal of ammonia in a wide range of pH, instead of the biological nitrification processes.



*2.5. Growth under Controlled Condition*

The cultivation of the strains was successfully done in an indoor laboratory at the Environmental Research Center, UiTM Dungun for 14 days using the low salinity water medium. The experiment was carried out in triplicate. Twelve batch cultures of both species were grown in 500 mL conical flasks with an initial cell concentration of 400 mg dry weight/L [18] in each 250 mL culture medium. All flasks were placed in a room at 25 ($\pm$ 2) °C and exposed to a white fluorescent light. The wavelength ranged from 480 to 570 nm, Philips (36 W) sources were placed 20 cm above the culture surface to ensure that all cells were equally exposed to the light with a 12/12 h light/dark photoperiod automatically controlled by a 24 h timer (Timer Bainian, BND-50/339, Cixi City, China) [19]. The light intensity was 325 Lux (32.5 mmol m$^2$ s$^{-1}$), measured from the culture surface using a light meter (model MS6612 Digital Lux Meter, Mastech Instruments, Azcapotzalco, Mexico D.F). The cultures were stirred with constant mixing using magnetic stirring bars (12 $\times$ 5 mm) at 40 rpm [18].

*2.6. Carbon Dioxide Treatment Condition*

A group of six batch cultures in 500 mL conical flasks (both species) were aerated by bubbling $CO_2$ at the volumetric flow rate of ambient air (0.03% of $CO_2$) controlled at 0.35 L min$^{-1}$ using calibrated mass flow controllers (model ST-51, FCI Flow meter, San Marcos, CA, USA). Then, another group of six batch cultures in 500 mL conical flasks (both species) were aerated in two stages, starting with ambient air (0.03% of $CO_2$) and enriched with 15% of $CO_2$. All flasks were covered with rubber caps and the aeration was introduced through an air filter of gas inline filter specifications (1 3/8″ or 35 mm diameter and 2 3/4″ or 67 mm long with a 60 μm filter element).

*2.7. Nitrate and Phosphate Ion Determination*

Nitrate and phosphate ion concentrations were measured during the 14 days cultivation period. A triplicate of water samples (10 mL) were collected daily. The determination of nitrate and phosphate ion concentrations was conducted using a portable UV-visible spectrophotometer (HACH DR900, Loveland, CO, USA). UV Screening Method 10049 and PhosVer 3 with Persulfate UV Oxidation1 Method 8007, Loveland, CO, USA, were used. The reagent of nitrate and phosphate ions was dissolved in 10 mL water samples to measure the concentration of water soluble $NO_3^-$ (nitrate ions) and $PO_4^{3-}$ (orthophosphate ions). The samples were centrifuged at 5000 rpm for 30 min.

*2.8. Statistical Analysis*

One-way analysis of variance (ANOVA) using Microsoft Excel 2010 data analysis was utilised to evaluate the experimental data. It was determined that all data had strong significance; the results from *p*-value < $2.190 \times 10^{-8}$. A small *p*-value (typically $\leq 0.05$) indicates strong evidence against the null hypothesis in the removal of nitrate and phosphate ions towards the cultivation period containing different types of microalgae strains [20].

**3. Results and Discussion**

Table 1 shows the characteristics of raw water from Sungai Sura nearby UiTM Dungun. All the parameters were determined according to the American Public Health Association (APHA) 1998 guidelines [21]. The characteristics of Sungai Sura downstream water sample were within the permissible discharge limit of National Water Quality Standards for Malaysia, classes IIA/IIB. The initial concentrations of nitrate and the phosphate ions in raw fresh water from Sungai Sura were in the range of 1.17 to 2.43 and 0.1 to 1.97 mg/L, respectively. After the addition of VermiCompost Tea Oganic Liquid Fertiliser, the concentrations of nitrate and phosphate ions increased to 5.96–5.27 and 0.52–2.07 mg/L, respectively.

**Table 1.** The characteristics of raw water from Sungai Sura nearby UiTM Dungun.

| Parameter | Sungai Sura Fresh Water | Sungai Sura Fresh Water Initial Concentration | Addition of Organic Fertiliser | Analysis Method | Discharge Limit (NWQSM) [a] |
|---|---|---|---|---|---|
| pH@25 °C | 6.60–7.18 | 7.18 | 7.00–8.00 | APHA 4500-H$^+$B | 6–9 |
| Dissolved oxygen (mg/L) | 6.23–7.29 | 6.23 | - | APHA 2520 B | 5–7 |
| Total suspended solid (mg/L) | 32–48 | 32 | - | APHA 2540 D | 50 |
| Nitrate ion, $NO_3^-$ (mg/L) | 1.71–2.43 | 1.71 | 5.96–5.27 | APHA4500-$NO_3$B | 7 |
| Phosphate ion, $PO_4^{3-}$ (mg/L) | 0.10–1.97 | 0.10 | 0.52–2.07 | APHA 4500 P B,C | 0.2 |
| Salinity (ppm) | 9054–9992 | 9054 | - | APHA 2520 B | - |

[a] National Water Quality Standards for Malaysia (NWQSM) class IIA/IIB.

Figure 1 shows the pH recorded for the 14 day cultivation period aerated with 0.03% (ambient air) and 15% of $CO_2$ for *Tetraspora* sp. and *Spirogyra* sp. The initial pH values were in the range of 7.08 to 7.18 and the final pH values were in the range of 6.18 to 6.80.

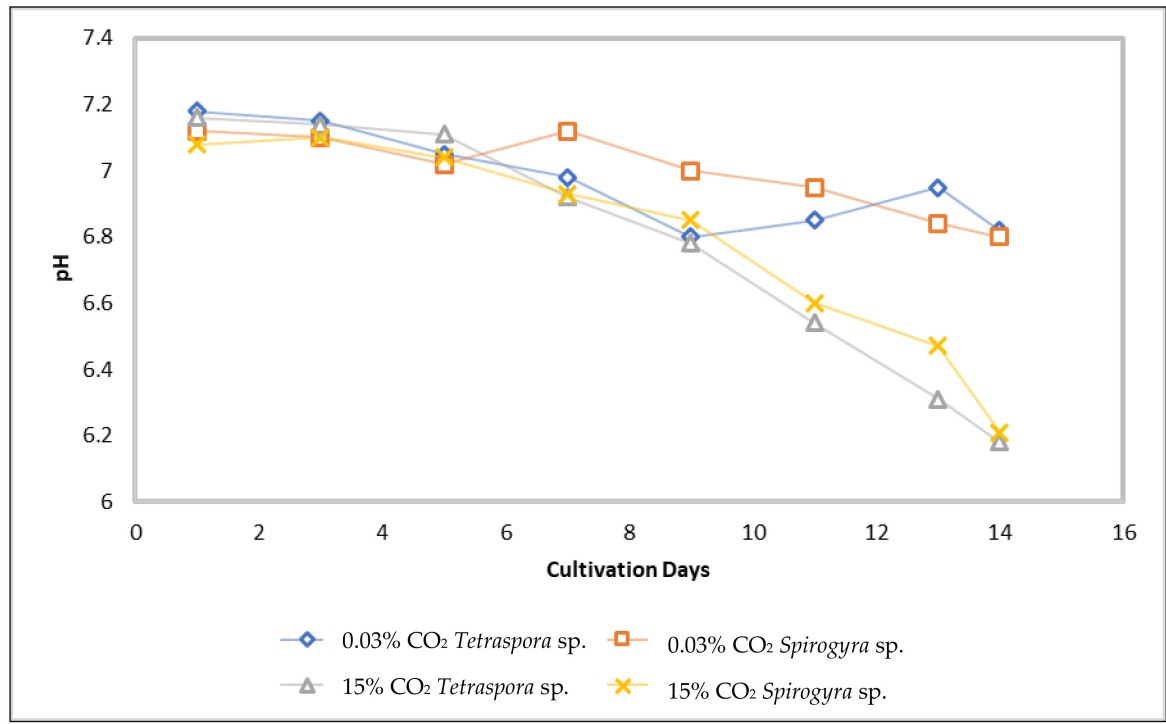

**Figure 1.** pH profiles during the 14 day cultivation of *Tetraspora* sp. and *Spirogyra* sp.

## 3.1. Effect of 0.03% of $CO_2$

### 3.1.1. Nitrate Ion ($NO_3^-$) Analysis

Figure 2 shows the concentration and reduction of nitrate ions for both microalgae species during the 14 day cultivation period. The initial nitrate ions concentration was $5.96 \pm 0.28$ mg/L and reduced to $0.37 \pm 0.05$ mg/L in *Tetraspora* sp. culture and $2.35 \pm 0.19$ mg/L reduced to $0.59 \pm 0.08$ mg/L in *Spirogyra* sp. culture aerated with 0.03% of CO2 (ambient air). The nitrate ions removal by *Tetraspora* sp. was better ($95.33 \pm 0.83\%$) than that by *Spirogyra* sp. ($84.02 \pm 2.16\%$) during the 14 day cultivation using Sungai Sura fresh water medium. The reduction of nitrate ions may be explained by the fact that both species consumed this inorganic substance as a macronutrient for growth and the photosynthesis process [6,22]. These results are consistent with the other research on the aqua cultural water ecosystem, that reported the application of the hybrid concept in removing nutrients and simultaneously cultivating microalgae species to produce biomass, which can be further exploited as live feed and a valuable biochemical resource of raw materials [23].

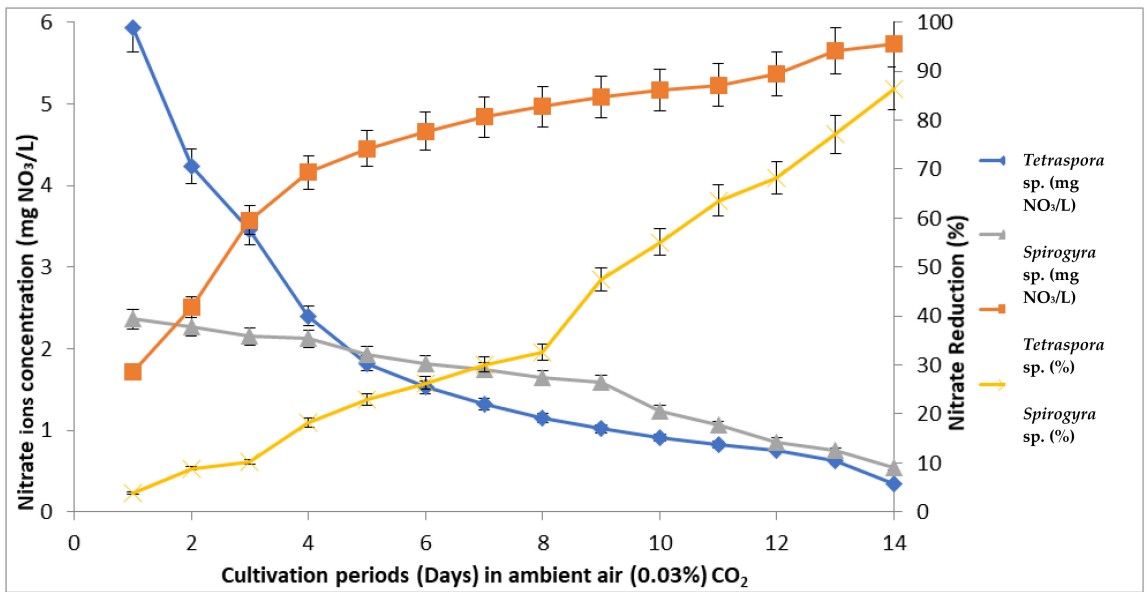

**Figure 2.** Nitrate ions ($NO_3^-$) concentration profile during the 14 day cultivation periods for *Tetraspora* sp. and *Spirogyra* sp. aerated with ambient air (0.03% of $CO_2$).

The main equations used to determine the nitrogen uptake are shown in Equations (1)–(3):

$$N_{balance} = N_{input} - N_{output} \tag{1}$$

$$N_{input} = (N_{fertiliser} + N_{water}) = (4.25 + 1.71) \text{ mg/L} = 5.96 \text{ mg/L} \tag{2}$$

$$N_{output} = N_{uptake \ by \ algae/removal} = (5.96 - 4.52) \text{ mg/L} = 1.44 \text{ mg/L} \tag{3}$$

### 3.1.2. Phosphate Ion ($PO_4^{3-}$) Analysis

Figure 3 shows the concentration and reduction of phosphate ions for both microalgae species during the 7 day *Spirogyra* sp. and 14 day *Tetraspora* sp. cultivation periods for *Tetraspora* sp. The initial phosphate ion concentration was $0.52 \pm 0.10$ mg/L and totally removed after 14 days by *Tetraspora* sp. In the *Spirogyra* sp. cultivation, the initial phosphate ions concentration was $0.63 \pm 0.15$ mg/L and totally removed after 6 days. Phosphate ions were totally removed in both cultures however, the duration for *Tetraspora* sp. was longer compared that of *Spirogyra* sp. These results are supported by the previous studies that reported the microalgae species are capable of utilising phosphate ions as the main nutrient for growth in water systems [24]. These results are likely to be related to the high concentration of phosphate ions exists in various water mediums which is beneficial for microalgae growth and at same time can treat the water medium [25]. The removal of phosphate ions and the treatment of water results in the generation of microalgal biomass that can be used as a feed stock in producing biodiesel [22]. The main equations used to determine the phosphate uptake are shown in Equations (4)–(6):

$$P_{balance} = P_{input} - P_{output} \tag{4}$$

$$P_{input} = (P_{fertiliser} + P_{water}) = (0.42 + 0.10) \text{ mg/L} = 0.52 \text{ mg/L} \tag{5}$$

$$P_{output} = P_{uptake \ by \ algae/removal} = (0.52 - 0.50) \text{ mg/L} = 0.02 \text{ mg/L} \tag{6}$$

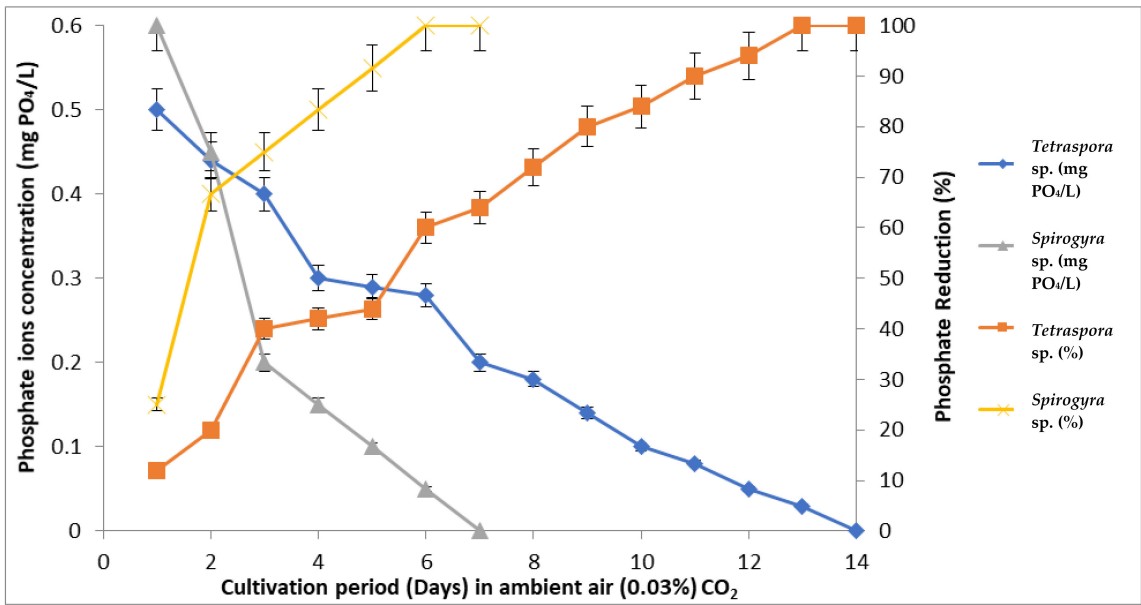

**Figure 3.** Phosphate ions ($PO_4^{3-}$) concentration profile during the 14 day cultivation period for *Tetraspora* sp. and *Spirogyra* sp.

### 3.2. Effect of 15% of $CO_2$

### 3.2.1. Nitrate Ions ($NO_3^-$) Analysis

Figure 4 shows the concentration and reduction of nitrate ions for both strains during the 14 day cultivation period. The initial nitrate ion concentration was $5.27 \pm 0.06$ mg/L and reduced to $1.80 \pm 0.20$ mg/L in *Tetraspora* sp. culture and $4.73 \pm 0.12$ mg/L reduced to $2.80 \pm 0.10$ mg/L in *Spirogyra* sp. culture, aerated with 15% of $CO_2$. The nitrate ions removal by *Tetraspora* sp. was better ($71.52 \pm 1.91$%) than that *Spirogyra* sp. ($64.07 \pm 4.26$%) during the 14 days cultivation using Sungai Sura fresh water medium.

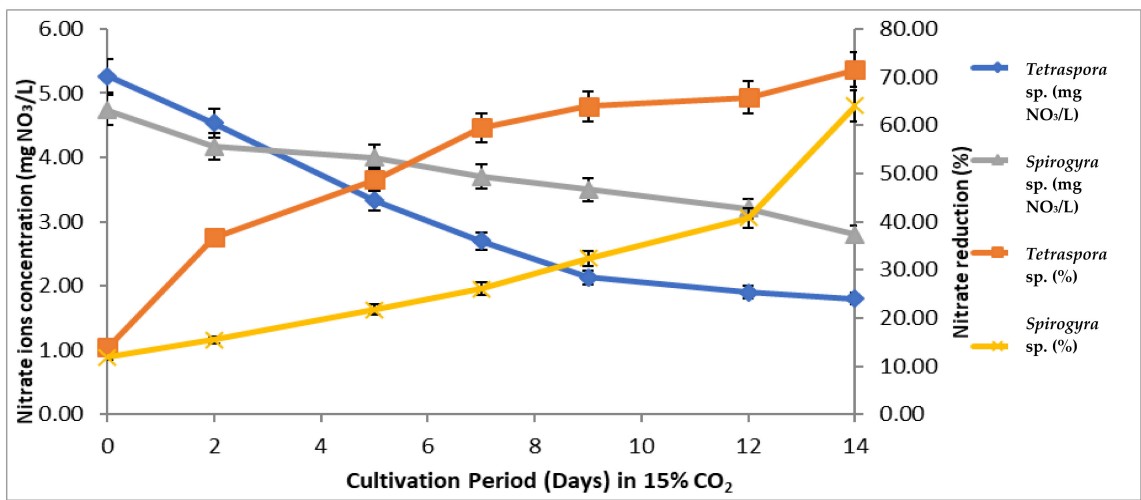

**Figure 4.** Nitrate ions ($NO_3^-$) concentration profile during the 14 day cultivation period for *Tetraspora* sp. and *Spirogyra* sp. aerated with 15% $CO_2$.

Carbon dioxide ($CO_2$) is an important compound in photosynthesis. However the high concentration of $CO_2$ will create harmful effects to growth and retard the growing process of microalgae where $CO_2$ is dissolved in an aqueous environment forming carbonic acid ($H_2CO_3$) [11]. Carbonic acid then dissociates to produce $H^+$ and bicarbonate ions ($HCO_3^-$). The culture medium will be acidic (decreased pH) due to the formation of a bicarbonate ions. Basically, $CO_2$ addition

and microalgae growth are closely interconnected to the carbon concentration mechanism (CCM). Increasing $CO_2$ concentration significantly inhibits the carbonic anhydrase (CA) activity which has a significant negative effect on microalgae cell formation. A low pH value may inhibit the activity of CA, which indicates an important catalytic role in the interchange between $CO_2$ and $HCO_3^-$. It is also considered as an important factor for CCM. As a result, the performance of carbon bio-fixation to numerous microalgae is weakened [26]. The following equation shows the formation of bicarbonate ions:

$$CO_{2(g)} + H_2O_{(l)} \rightleftarrows H_2CO_3 \rightleftarrows H^+ + HCO_3^-$$

Futhermore, due to the CCM factor, the reduction of microalgae growth is closely related to the low consumption of nitrate ions. The aeration of 15% of $CO_2$ leads to excess $CO_2$ that is converted into $H_2CO_3$ and hence reduces the pH of the medium, consecutively influencing algae growth [27]. Figure 1 shows a decrease in pH values on days 12 to 14 due to the formation of carbonic acid ($HCO_3^-$).

Numerous studies on $CO_2$ capture reported that a flow rate of 0.25 vvm or 2% (*v/v*) of $CO_2$ is the optimum for the growth of algae species of chlorophycean, while at a flow rate of 10% (*v/v*) of $CO_2$ makes the specific growth rate insignificant. The increase in $CO_2$ concentration contributes to an impediment effect [28] on the growth of *Chlorella sorokiniana* [26].

### 3.2.2. Phosphate Ion ($PO_4^{3-}$) Analysis

Figure 5 shows the concentration and reduction of phosphate ions for both microalgae species during the 14 day cultivation. The initial stage concentration of phosphate ions was $0.57 \pm 0.15$ mg/L and reduced to $0.13 \pm 0.06$ mg/L in *Tetraspora* sp. Culture, and $0.32 \pm 0.08$ mg/L reduced to $0.10 \pm 0.10$ mg/L in *Spirogyra* sp. culture aerated with 15% of $CO_2$.

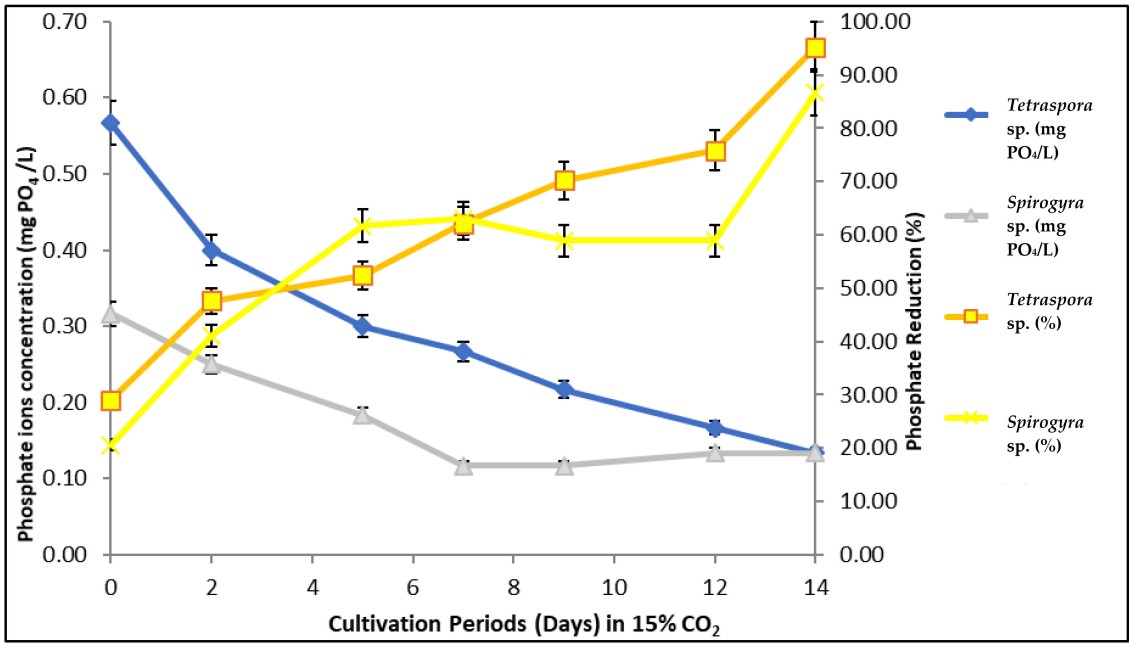

**Figure 5.** Phosphate ions ($PO_4^{3-}$) concentration profile during the 14 days cultivation for *Tetraspora* sp. and *Spirogyra* sp. aerated with 15% of $CO_2$.

These results indicated that the phosphate ions removal in *Tetraspora* sp. culture was better than that by *Spirogyra* sp. culture during the 14 day cultivation using the Sungai Sura fresh water medium. Many studies demonstrated the success of using algal cultures to remove nutrients from wastewater rich in nitrogenous and phosphorus compounds. Another chlorophycean species *Scenedesmus* sp. is very common in all kinds of fresh water bodies, which play an important role as a primary producer

and contributes to the purification of eutrophic waters [6]. Microalgae are particularly attractive in the bio-treatment of waste-water because of their photosynthetic capability of converting solar energy into useful biomass. Microalgae also consist of nutrients such as nitrogen and phosphorus which leads to the eutrophication process. Removal of nutrients such as nitrate and phosphate ions are directly linked to photosynthetic activity and biomass production [29].

## 4. Comparing Other Studies in Reduction of Nitrate and Phosphate Ions

*Platymonas subcordiformis* grown in natural seawater yielded in 87–95% nitrate ion reduction during a 14 day cultivation period [4,30]. Meanwhile, *Chlorella* sp. grown in wastewater gave 84.11% nitrate ion reduction during a 14 day cultivation period [31]. The results of 95.33 ± 0.83% and 84.02 ± 2.16% nitrate ion removals in *Tetraspora* sp. and *Spirogyra* sp. culture after the 14 day cultivation in Sungai Sura fresh water medium is comparable to those reported in other studies using different culture mediums. All the results indicated a negative correlation (reduction of nitrate and phosphate ions) of *Tetraspora* sp. and *Spirogyra* sp. against cultivation days. According to the literature, microalgae cell consumption of nitrate ions for growth can be removed from water and transmission to the microalgae cell through bioadsorption and storage inside the cell (bioaccumulation) [32].

Phosphate ion reduction by algae species depends on the different types of medium such as *Platymonas subcordiformis* grown in natural seawater (98–99% reduction) [30] and *Chlorella* sp. grown in wastewater (82.36% reduction) [31]. Although nitrate ions are an important nutrient for plant growth, another main element required for the growth of algae is phosphorus [6]. Natural waters usually contain excess concentration of organic phosphate ions which is important to be in soluble form to be acquired by the microalgae cell [13]. The application of algae for the mitigation of phosphate ions, nitrate ions and ammonia (also known as biological nutrient removal) in wastewater treatment has been the subject of study since around the mid-1970's which provides an economical and environmentally sustainable alternative treatment method [33].

## 5. Growth of Microalgae and Fatty Acid Methyl Ester (FAME)

According to Rozaimee et. al. (2016) [18], the maximum growth of microalgae species of *Tetraspora* sp. and *Spirogyra* sp. in low salinity water was observed on the twentieth (20) day (densities of *Tetraspora* sp. were 7.37 mg/mL and *Spirogyra* sp. were 3.16 mg/mL). In a related study, 62.22% and 30.73% of *Tetraspora* sp. and *Spirogyra* sp. lipids were respectively converted to fatty acid methyl esters (FAME) [34]. These outcomes indicated that both microalgae species have the potential for the production of biofuel in the future.

## 6. Conclusions

Based on this study, microalgae shows great potential as a sustainable alternative aqua plant for the removal of organic nutrients such as nitrate and phosphorus ions from Sungai Sura water. The cultivation of both microalgae strains aerated with ambient air (0.03% of $CO_2$) decreased 95.33 ± 0.83% of nitrate ions by *Tetraspora* sp. and 84.02 ± 2.16% by *Spirogyra* sp. After a 14 day cultivation processes. Meanwhile, 100% removals of phosphate ions were achieved after 14 days for *Tetraspora* sp. cultures and 7 days for *Spirogyra* sp. cultures. However, the cultivation of both microalgae strains with the aeration of 15% of $CO_2$ decreased the nitrate ion removal to 71.52 ± 1.91% for *Tetraspora* sp. and 64.07 ± 4.26% in *Spirogyra* sp. After the 14 day cultivation process, due to the excessive $CO_2$ in the water medium, changed the pH (acidity). The results of this research showed successful reduction of a higher content of nitrate and phosphate ions through biological treatment (microalgae cultivation), specifically *Tetraspora* sp. and *Spirogyra* sp. cultivation in ambient air (0.03% of $CO_2$), in a fresh water medium. Meanwhile, the aeration of 15% of $CO_2$ retarded the removal of nitrate and phosphate ions due to the formation of carbonic acid ion ($HCO_3^-$) in the water medium that also lowered the biomass production.

In addition, microalgae-based biomass is one of the most promising feedstocks for the production of biofuel such as bioethanol, biogas, and biodiesel due to their promising sustainable advantages, along with their high productivity compared to traditional earthly feedstock.

**Author Contributions:** Investigation, A.R.M. and M.M.; Supervision, K.H.K.H. and J.I.; Writing—original draft, A.R.M., R.R. and J.I.; Writing—review and editing, A.R.M. and J.I.

**Funding:** This research received no external funding.

**Acknowledgments:** We would like to thank the Universiti Teknologi MARA (UiTM), Bukit Besi Campus and Sura Hujung Campus, Terengganu for providing the necessary facilities to carry out this research and Makmal Penyelidikan Alam Sekitar, Faculty of Chemical Engineering, Shah Alam for the support and cooperation. Last but not least, we are also a grateful to the Ministry of Higher Education, Malaysia for the financial support of the Doctorate Scholarship Programme 2013.

**Conflicts of Interest:** The authors declare no conflict of interest.

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
