# Peer review of "High Nitrate and Phosphate Ions Reduction in Modified Low Salinity Fresh Water through Microalgae Cultivation"

_processes, doi:10.3390/pr7030129_

Round 1
Reviewer 1 Report
General comment:
The paper deals with investigations on the reduction of nitrate and phosphate in modified low salinity surface water via cultivating microalgae strain e.g Tetraspora sp. And Spirogyra sp. Microalgae cultivation was carried out under controlled condition in indoor laboratory for 14 days. Investigations were carried out both with ambient air and with enhanced air (15% CO2).
The manuscript is suitable to be published in this journal; however, some major points should be addressed before publication.
There are several vagueness that have to be solved. Moreover, please, read carefully the manuscript because there are some incongruences, for example page 8, lines 248-251 - discussion is about phosphate and not nitrate.
Some minor language mistakes are present that should anyway be corrected.
1. Introduction
Please, improve this section by including a short overview on by-product production from microalgae. Please, consider the following papers:
o Molino, A., Rimauro, J., Casella, P., Cerbone, A., Larocca, V., Chianese, S., Karatza, D., Mehariya, S., Ferraro, A., Hristoforou, E., Musmarra, D., 2018. Extraction of astaxanthin from microalga Haematococcus pluvialis in red phase by using generally recognized as safe solvents and accelerated extraction, Journal of Biotechnology, 283, 51-61.
o Damergi, E., Schwitzguébel, J.-P., Refardt, D., Sharma, S., Holliger, C., Ludwig, C., 2017. Extraction of carotenoids from Chlorella vulgaris using green solvents and syngas production from residual biomass, Algal Research, 25, 488-495.
o Di Sanzo, G., Mehariya, S., Martino, M., Larocca, V., Casella, P., Chianese, S., Musmarra, D., Balducchi, R., Molino, A., 2018. Supercritical carbon dioxide extraction of astaxanthin, lutein, and fatty acids from haematococcus pluvialis microalgae, Marine Drugs, 16(9), Article number 334.
o Albarelli, J.Q., Santos, D.T., Ensinas, A.V., Maréchal, F., Cocero, M.J., Meireles, M.A.A., 2018. Comparison of extraction techniques for product diversification in a supercritical water gasification-based sugarcane-wet microalgae biorefinery: Thermoeconomic and environmental analysis, Journal of Cleaner Production, 201, 697-705.
Futhermore, the possibility of use microalgae to produce biofuel should be stressed to outline the use of microalgae produced by using polluted medium; have a look to the recent papers:
o Sajjadi, B., Chen, W.-Y., Raman, A.A.A., Ibrahim, S., 2018. Microalgae lipid and biomass for biofuel production: A comprehensive review on lipid enhancement strategies and their effects on fatty acid composition, Renewable and Sustainable Energy Reviews, 97, pp 200-232.
o Molino, A., Larocca, V., Chianese, S. and Musmarra, D., 2018. Biofuels production by biomass gasification: a review, Energies, 11(4), 811.
2.2. Sampling Raw Water Method
Please, include raw water characterization, in particular highlighting the initial concentration of nitrate and phosphate and salinity.
2.3. Addition of Organic Fertilizer
Please, include fertilizer characterization.
2.4. Ozone Treatment of Culture Medium Preparation
Please, include the characterization of culture medium after ozone treatment, in particular highlighting the initial concentration of nitrate and phosphate and salinity.
2.5. Growth under controlled condition
Please, specify the wavelength of the light.
Please, include microalgae growth measurement.
3. Results and Discussion
In my opinion Table 1 should be reported in section 2, where the materials with their characteristics are reported
Please, check numeration of Tables.
3.1.1. Nitrate (NO3) analysis
Figure 1: please, improve readability.
Please, compare experimental findings with literature data in terms of rate of nitrogen used by microalgae for their growth..
Please, include nitrogen material balance.
3.1.2. Phosphate (PO43-) analysis
Please, specify the meaning of “nil” in Table 2.
Figure 2: please, improve readability.
According to Figure 2, it seems that a removal of phosphate higher than 100% was achieved: please verify.
Please, compare experimental findings with literature data in terms of rate of phosphate used by microalgae for their growth.
Please, include phosphate material balance.
3.2. Effect of 15% CO2
The transfer of CO2 from the gaseous phase to the liquid phase should be estimated.
Please, include carbon material balance.
3.2.1. Nitrate (NO3) analysis
Table 3: please, check “Culture day” column because it is not possible that you found a phosphate removal at Day 0.
3.2.2. Phosphate (PO43-) analysis
Figure 3: if the phosphate removal is 100% why the residual concentration is not zero? Please, check data.
Table 4: please, check “Culture day” column because it is not possible that you found a phosphate removal at Day 0.
Author Response
Dear Reviewer
Attached herewith edited manuscript based on your comments and reviewer's response (point by point) which can be found at the end of manuscript ( after references)
thank you
regards
juferi

Reviewer 2 Report
Overall this paper is well written. The experiment is well designed and method is solid. The results are well presented and discussed. I believe this paper can be further improved. Below are my comments:
The introduction gives no information about the novelty of the work.
In the abstract, provide the implication of the results achieved
There are a couple of grammatical errors that need to be corrected e.g. check lines 39, 44-45, 106 etc.
Author Response

(The authors gave the same response as above.)

Reviewer 3 Report
The manuscript “High nitrate and phosphate reduction in modified low salinity surface water through microalgae cultivation” describes and analyzes the reduction of nitrate and phosphate in low salinity surface water due to the action of microalgae species Tetrospora and Spirogira under two different CO2 concentration (0.03% and 15%). The paper is scientifically sound and interesting, and it would improve the knowledge of the research community on the topic. Before being accepted for publication in Processes, its quality needs to be improved based on reviewers’ comments and remarks; in particular, English style and grammar MUST be completely revised and dramatically improved. Other relevant issues are present, as listed below.
1) Please revise English style and grammar, as it is extremely hard to read and understand your manuscript.
2) Data are presented two times, both in table and figure forms (e.g., Fig. 1 and Table 1; Fig.2 and table 2; Fig. 3 and Table 3; Table 4 and Fig. 4).
3) More discussion should be added in each subsection, and a deeper discussion about the mechanisms of nitrate and phosphate removal should be implemented. A deeper comparison with other works in literature dealing with the same species should be introduced.
4) Did the authors measured NO2- or Total Nitrogen concentration? In order to assess the success of denitrification, the identification of intermediate N-forms is necessary, as they are toxic and/or GHG.
5) In section 2.8. there is the description of the statistical analysis performed, but then in the text there is no mention of the results. Why a p-value of 2.19*10-8 was chosen? Please provide a reference.
6) Having a look at the NO3- removal graphs, it looks that the removal of nitrate by Spirogira is linear, while that of Tetraspora may be well descripted by a 2nd or 3rd order polynomial. I would suggest to investigate the fact also in the case of phosphate removal, as a simple but effective regression model could be easily implemented in both cases, and would greatly improve the quality of your manuscript. Did this trend is reported in other literature?
7) Citation style needs to be consistent with journal guidelines.
8) L52: please add a reference
9) L37-40: Even if it is a general introductory section, please use references.
10) L65: This is a strong sentence, please provide a reference.
11) Please provide composition and producer of VermiCompost Tea.
12) Table 2: what is “nil”?
13) L224: a reference is needed.
Author Response

(The authors gave the same response as above.)

Reviewer 4 Report
Comments on the Graphical abstract: No graphical abstract
Comments on the highlights: No highlight.
Keywords: May be freshwater would be more adequate than low salinity. You should add Spirogyra and Tetraspora.
Global comments on the paper:
The paper presents a cultivation of two phytoplankton species under artificially nutrient enriched freshwater. The aim is to test the feasibility to use these two microalgae to fix nitrates and phosphates from water, then to produce bioenergy. The objective is interesting. Unfortunately, the paper suffers from several insufficiencies. Please refer to my hereafter details as main issues to solve:
Global comment: the English must be improved. A lot of sentences are approximative with wrong wordings, or are confusing.
Lines 23-26: Why didn’t you consider a same initial water quality for each species and in each case? It leads to a much more complex diagnosis of your results.
Lines 26-28: This sentence is not clear. It seems to refer to the same aspect as the preceding sentence, whereas providing different results. You should rephrase your sentence for a better understanding of the abstract.
Line 48: Eutrophication is not the direct consequence of oxygen depletion. It is the reverse, eutrophication leads to oxygen depletion, and oxygen depletion can indirectly sustain eutrophication. Please replace "results in" by "sustains"
Line 53: please replace “many” by “Several”. Indeed, a lot of phytoplankton species are not able to efficiently grow in wastewater.
Line 56: replace “Abdel-Raouf, Al-Homaidan, & Ibraheem” by “Abdel-Raouf et al”. Moreover, it should be better to add specific references in temperate waters (your reference is from Saudi Arabia).
Lines 56-60: your sentence must be revised as it is inconsistent, mixing parts of several sentences with technical errors. For instance, residential septic tanks have generally a low level of nitrate and a high level of ammonium and organic nitrogen. Moreover, detergents are not from agriculture. Hence I don’t understand how detergents could encourage microalgal growth!!
Line 60: replace “(Jiang, Luo, Fan, Yang, & Guo, 2011) (Ponnuswamy, Madhavan, & Shabudeen, 2013” by “(Jiang et al, 2011; Ponnuswamy et al, 2013)”
Line 62: you have forgotten that the main option consist of treating wastewaters prior to their arrival in rivers.
Lines 64-65: you are wrong. The main microorganisms used in wastewater treatment are bacteria. Do you imagine that microalgae that produce oxygen can be used in anaerobic ponds?!
Line 66: Do you imagine that the use of cyanobacteria is an environmentally friendly wastewater treatment?!
Lines 69 and 72: replace “Maity, Bundschuh, Chen, & Bhattacharya, 2014a” and “Maity, Bundschuh, Chen, & Bhattacharya, 2014b” by “Maity et al, 2014a” and “Maity et al, 2014b”
Line 75: what is Sungai Sura ? Please provide more detail on that (also refer to my comment 15)
Lines 75-76: Could you justify the test of enriched CO2 on algal growth. Please provide some comments on CO2 theoretical impact on algal growth.
Lines 84-86: Please provide more details on this river. Is it downstream a huge watershed, is it contaminated by urban wastewaters…?
Lines 89-91: could you confirm you use twice a 500ml to fill a 1500ml bottle with 1000ml of water. It means you have still 500ml of air inside this bottle. Did you collect these two raw water samples at the same site and mix them in a single 1000ml sample, or you collect two 1000ml samples at 2 different (or two identical) sites??? Please be so kind to clarify.
Line 91: Why did you mention 4°C line 85 if you practically stored the samples at 10-15°C?
Line 92: did you analyze DO and pH in the river, or at the arrival in the laboratory? It is practically different.
Line 92: Your list of surveyed parameters is not sufficient. You lack considering key parameters, such as organic matter, organic nitrogen, total phosphorus. It is a strong weakness of your methodology, because you should precisely know what is the total balance of nitrogen and phosphorus in your raw water samples prior to the laboratory tests.
Line 94: You mention that you have 3000ml whereas you indicate line 90 that your samples are 1000ml. Please could you clarify.
Line 95: Please provide the exact values for these parameters in the raw water.
Line 96-98: What is the composition of this fertilizer? You must provide the values for nitrates and phosphates, but also for the total nitrogen and total phosphorus. Indeed, you indicate that your fertilizer includes organic plant food.
Lines 96-98 and Lines 108-110: You should not avoid the potential mineralization of organic N and P in your laboratory study. Practically, you don’t know what is the real nitrogen and phosphorus concentrations at the beginning of your tests. It is a strong issue and weakness of your methodology.
Line 105-106: You mention a “low salinity water due to the river intercept with sea water”. If you collect water in the estuary, it is probable that the water salinity is not low”. Could you clarify what do you mean.
Line 107: Please provide the ozone dosing and contact time.
Line 107: Please provide more details on your filtration process.
Line 118: What do you mean by 1ml of algae? Please provide the concentration in cells/ml.
Line 122: Please replace “Taher, Al-Zuhair, Al-Marzouqi, Haik, & Farid” by “Taher et al”
Line 125-126 : Please replace « Rozaimee Mustaffa, Halim Ku Hamid, Musa, Salihon, & Ramli” by “Rozaimee et al”
Line 138-139: If you collected 20ml per day for 14 days, it means that at the end of the process, the water was reduced by 280ml (starting from 500ml), meaning a 220ml remaining in the water samples. How did you consider that water volume decrease in your test (for instance regarding the air bubbling), and in the results diagnosis?
Line 142-143: You should have also analyzed total phosphorus and global nitrogen.
Line 151 - Table 1: why did you measure pH at 25°C whereas your cultures were at 27°C (line 119)?
Line 151 - Table 1: why didn't you use raw samples with the same water quality for all your tests?
Line 151 - Table 1: please provide the phosphate unit. It is not sufficient to indicate “mg/l”.
Line 151 - Table 1: You have a difference of phosphate concentration up to 2mg/l between your raw water qualities, whereas you have only a limited variation of phosphate concentration after adding of fertilizers? Could you clarify this aspect.
Line 151 – Table 1: You should have measured a lot of other parameters to qualify the other chemical parameters in your raw water and during the tests. How could you be sure that no parameter was limiting for algal growth?
Line 154: your reference to APHA (1998) is not in your reference list.
Line 158: “w” to suppress.
Line 164 -Table 1: it should be table 2.
Line 164 – Table 2: please provide results in cells/ml. Or if you use phytoplankton in mg/l be so kind to explain in your methodology how you measure that. Is it a wet weight, a dry weight? What is the biovolume you use for each species cell?
Line 179: replace “Hii, Soo, & Chuah » by « Hii et al ».
Line 181: Please comment the difference in the shape of the curves for the 2 different microalgal species.
Line 181: please indicate the nitrate unit on the figure. Here again it is not sufficient to indicate mg/l
Line 187: table 2 is practically table 3.
Line 193-195: inconsistent wording.
Line 195-197: your sentence is wrong. Bhatt et al (2014) have not found that “microalgae species require phosphate as a nutrient for growing in water”. This was known for more than 40 years. ¨Please modify your sentence.
Line 197: replace “Bhatt, Panwar, Bisht, & Tamta” by “Bhatt et al” if you keep that sentence some place in your paper.
Line 198-199: replace “Rashid, Ur Rehman, Sadiq, Mahmood, & Han” by “Rashid et al”
Line 199-201: your sentence is inconsistent. Please revise it.
Line 202: your figure 2 is inconsistent with your table 3. How could you have more than 100% reduction?
Line 202: please add the phosphates unit, and replace “(mg/l” by (mg/l).
Line 207-209: table 3 is table 4.
Line 209: why didn’t you use the same raw water quality for starting your tests with the 2 different CO2 concentrations? It could have an impact on the results, then on the diagnosis.
Line 209: please indicate the nitrate unit on the figure.
Line 218: please indicate the nitrate unit on the figure.
Line 225: provide the daily variation of pH during the tests. It is essential to demonstrate that your hypothesis is valid.
Line 231: You cannot mention that for all the microalgae. It depends of the species.
Line 237: please replace “Kaewkannetra, Enmak, & Chiu” by “Kaewkannetra et al”.
Line 237: you refer to a paper focusing on a scenedesmus species that is different than your two species. Refer to my above-mentioned comment 57.
Line 241: replace “Millán-Oropeza, Torres-Bustillos, & Fernández-Linares » by « Millán-Oropeza et al »
Line 246: table 4 is table 5.
Line 246: indicate the phosphate unit.
Line 246: how could you indicate a 100% reduction whereas phosphate values are still higher than 0 ?!
Line 253: your figure 3 title is wrong. It is phosphate not nitrate.
Line 259: you should justify the choice of your 2 species. What are their specificities? Why didn’t you use other chlorophycean species such as scenedesmus?
Line 265: replace “Judd, van den Broeke, Shurair, Kuti, & Znad » by « Judd et al”
Line 268: table 3 is table 4.
Line 270: replace “Littler et al., 2006)(Guo, Liu, Guo, Yan, & Mu, 2013)” by “Littler et al, 2006: Guo et al, 2013)”
Line 274: “Rasoul-amini,et al »
Line 276 : table 5 is table 6.
Line 284 : Abdel-Raouf et al., 2012 are not the founder of phosphorus as key element for algal growth.
Line 287: replace organic phosphate by organic phosphorus. Phosphates are mineral.
Lines 267-289: Part 3.3 should be replaced by a part 4 Discussion. In this part, you need to include a more in-depth analysis of your results.
Lines 290-302: Conclusion: The sentences must be revised (approximative English). Moreover, you should avoid considering conclusion as a summary of your results. Conclusion is supposed to be a global overview with potential perspective.
Global comment: you should justify the choice of the two species you worked on. You need to clarify their potential for production of bioenergy.
Author Response

(The authors gave the same response as above.)

Round 2
Reviewer 1 Report
The authors revised the manuscript according to my suggestion. The manuscript can be published.
Author Response
Thank you for you kind and meaningful comments and suggestion
Reviewer 4 Report
Comments on the Graphical abstract: ok
Comments on the highlights: ok but grammar to revise.
Keywords: Ok but verify that you fulfil the maximum number of keywords.
Global comments on the paper:
The paper presents a cultivation of two phytoplankton species under artificially nutrient enriched freshwater. The aim is to test the feasibility to use these two microalgae to fix nitrates and phosphates from water, then to produce bioenergy. The objective is interesting.
Unfortunately, after revision 1, the paper still suffers from key insufficiencies. Please refer to my hereafter details as main issues to solve (I refer to the same question number as in my first assessment of the paper, and the requests from my first assessment are in bold when they have not been considered). Be also kind to refer to the line number that are in the left margin of the paper, as it will be much more easy to verify your answer:
Global comment- key issue: the English must be improved. A lot of sentences are approximative with wrong wordings, or are confusing. This issue is still essential to solve. The authors must be helped to revise the paper, as their English is still not sufficient.
Lines 23-26: Why didn’t you consider a same initial water quality for each species and in each case? It leads to a much more complex diagnosis of your results. This request has not been addressed. It is a key issue. It must at least be clarified in the paper. Practically the initial conditions were different for the two species that is not logical.
Lines 26-28: This sentence is not clear. It seems to refer to the same aspect as the preceding sentence, whereas providing different results. You should rephrase your sentence for a better understanding of the abstract. Did you address that request? You have not included this request in your answer to reviewer.
Line 48: Eutrophication is not the direct consequence of oxygen depletion. It is the reverse, eutrophication leads to oxygen depletion, and oxygen depletion can indirectly sustain eutrophication. Please replace "results in" by "sustains". Your sentence is still unclear and must be revised.
Line 53: Ok.
Line 56: replace “Abdel-Raouf, Al-Homaidan, & Ibraheem” by “Abdel-Raouf et al”. This part of the request is still to be performed.
Lines 56-60: OK
Line 60: OK
Line 62: you have forgotten that the main option consist of treating wastewaters prior to their arrival in rivers. I don’t find the sentence. Please be so kind to indicate the true line number.
Lines 64-65: you are wrong. The main microorganisms used in wastewater treatment are bacteria. Do you imagine that microalgae that produce oxygen can be used in anaerobic ponds?! This issue is lines 83-84 of your revised paper. You are still wrong. Microalgae are not used in anaerobic ponds.
Line 66: OK
Lines 69 and 72: Ok”
Line 75: OK
Lines 75-76: OK.
Lines 84-86: Please provide more details on this river. Is it downstream a huge watershed, is it contaminated by urban wastewaters…?. It is line 101 of your revised paper. You have not provided the requested details. It has still to be included in your paper.
Lines 89-91: Ok.
Line 91: Why did you mention 4°C line 85 if you practically stored the samples at 10-15°C? This is a good example of wrong wording. You mention the use of a freezer for storing sampled at 10-15°C. A freezer is under 0°C. It is inconsistent.
Line 92: did you analyze DO and pH in the river, or at the arrival in the laboratory? It is practically different. Here again the wording is not clear. You refer to your table 1. But this table is not totally explicit regarding where the analyses where performed.
Line 92: Your list of surveyed parameters is not sufficient. You lack considering key parameters, such as organic matter, organic nitrogen, total phosphorus. It is a strong weakness of your methodology, because you should precisely know what is the total balance of nitrogen and phosphorus in your raw water samples prior to the laboratory tests. I don’t think you have understood my comment. You have provided no answer to my request.
Line 94: OK.
Line 95: Please provide the exact values for these parameters in the raw water. There are several issues to solve in you table 1. You need to add the unit for phosphate and nitrates (it seems that you have not understood my different comments on that – phosphate can be expressed under different unit, all being in mg/l and it is the same for nitrates). Moreover, you refer to salinity in µS/cm whereas it is not a salinity unit but a conductivity unit. Please revise your table.
Line 96-98: What is the composition of this fertilizer? You must provide the values for nitrates and phosphates, but also for the total nitrogen and total phosphorus. Indeed, you indicate that your fertilizer includes organic plant food. You have not answered on the organic part of N and P in your fertilizer. This is a key issue as you modified the quality of the raw water. Moreover, you indicate that the concentration of inorganic P and N varied in the added fertilizer within a range. It means that you have not a clear view of the added P and N in your samples. Here again, it is an issue.
Lines 96-98 and Lines 108-110: You should not avoid the potential mineralization of organic N and P in your laboratory study. Practically, you don’t know what is the real nitrogen and phosphorus concentrations at the beginning of your tests. It is a strong issue and weakness of your methodology. I have seen your answer, but you should at least comment that issue in your paper.
Line 105-106: You mention a “low salinity water due to the river intercept with sea water”. If you collect water in the estuary, it is probable that the water salinity is not low”. Could you clarify what do you mean. You should precise line 103 that the collected water is freshwater.
Line 107: Please provide the ozone dosing and contact time. 25g/m3 is not a flow but a concentration.
Line 107: Please provide more details on your filtration process. Lines 172-173 of your revised paper are not totally explicit on that request.
Line 118: What do you mean by 1ml of algae? Please provide the concentration in cells/ml. 1ml of algae is not equivalent to 103 cells/ml. Please clarify what do you mean.
Line 122: Ok
Line 125-126 : Ok
Line 138-139: If you collected 20ml per day for 14 days, it means that at the end of the process, the water was reduced by 280ml (starting from 500ml), meaning a 220ml remaining in the water samples. How did you consider that water volume decrease in your test (for instance regarding the air bubbling), and in the results diagnosis?. You don’t answer the question. I understand that you have not adapted your bubbling to the remaining volume of water sample. Could you confirm what potential impact it could have.
Line 142-143: You should have also analyzed total phosphorus and global nitrogen. Yes it is a weakness that you should at least comment in your paper.
Line 151 - Ok
Line 151 - Table 1: why didn't you use raw samples with the same water quality for all your tests? You have not answered to my question. Do you have understood the request?
Line 151 - Table 1: please provide the phosphate unit. It is not sufficient to indicate “mg/l”. Here again you have not answered to the request. Do you have understood the request? You must indicate the form of the P and N in the corresponding units.
Line 151 - Table 1: You have a difference of phosphate concentration up to 2mg/l between your raw water qualities, whereas you have only a limited variation of phosphate concentration after adding of fertilizers? Could you clarify this aspect. You don’t answer the question.
Line 151 – Table 1: You should have measured a lot of other parameters to qualify the other chemical parameters in your raw water and during the tests. How could you be sure that no parameter was limiting for algal growth? You are wrong. You must clarify the potential toxicity of your water sample to be sure your test is not biased by existing toxicants. You should at least comment that issue in your paper.
Line 154: ok.
Line 158: ok.
Line 164 -Table 2 suppressed !!.
Line 164 – Table 2: please provide results in cells/ml. Or if you use phytoplankton in mg/l be so kind to explain in your methodology how you measure that. Is it a wet weight, a dry weight? What is the biovolume you use for each species cell? I don’t understand your answer. What is your unit of algae expressed in mg/l ?
Line 179: OK
Line 181: Please comment the difference in the shape of the curves for the 2 different microalgal species. You don’t comment the difference between the two curves.
Line 181: please indicate the nitrate unit on the figure. Here again it is not sufficient to indicate mg/l It has still to be performed. You have not addressed the issue.
Line 187: table 3 suppressed !!.
Line 193-195: inconsistent wording. English has still to be revised.
Line 195-197: OK
Line 197: OK.
Line 198-199: Ok
Line 199-201: your sentence is inconsistent. Please revise it. English has still to be revised.
Line 202: Table suppressed !!
Line 202: Ok
Line 207-209: table suppressed !!
Line 209: why didn’t you use the same raw water quality for starting your tests with the 2 different CO2 concentrations? It could have an impact on the results, then on the diagnosis. You don’t answer the question.
Line 209: please indicate the nitrate unit on the figure. You don’t answer the question.
Line 218: please indicate the nitrate unit on the figure. You don’t answer the question.
Line 225: provide the daily variation of pH during the tests. It is essential to demonstrate that your hypothesis is valid. You don’t answer the question. If you measured pH, you should be able to provide the results. Be so kind to add them.
Line 231: OK.
Line 237: Ok
Line 237: Ok.
Line 241: Ok
Line 246: table suppressed !!.
Line 246: indicate the phosphate unit. You have not answered the question.
Line 246: how could you indicate a 100% reduction whereas phosphate values are still higher than 0 ?! Figure 4 is still inconsistent.
Line 253: OK.
Line 259: you should justify the choice of your 2 species. What are their specificities? Why didn’t you use other chlorophycean species such as scenedesmus? You have not answered the request.
Line 265: Ok
Line 268: table removed !!.
Line 270: ok
Line 274: ok
Line 276 : table removed !!.
Line 284 : Abdel-Raouf et al., 2012 are not the founder of phosphorus as key element for algal growth. My comment is still valid. This was discovered more than 50 years ago, then not in 2018 nor in 2012.
Line 287: ok.
Lines 267-289: Discussion. In this part, you need to include a more in-depth analysis of your results. This request is still valid.
Lines 290-302: Conclusion: The sentences must be revised (approximative English). Moreover, you should avoid considering conclusion as a summary of your results. Conclusion is supposed to be a global overview with potential perspective. This request is still valid regarding the English.
Global comment: you should justify the choice of the two species you worked on. You need to clarify their potential for production of bioenergy. OK
Additional comment - Line 114: “the sample of freshwater medium (3 liters)” If you collect two samples of 1500ml as you indicate (lines 107-112) then mix these two samples in a single 3000ml sample, isn't it the same as if you have directly collected one 3000ml sample?
Additional comment - Line 130: replace salinity by conductivity
Additional comment - Line 178 of your revised paper: your part 2 “results and discussion” is part 3.
Additional global comment – main requests: Most of the main issues are still to be solved. Could you confirm if you clearly understand the requests/comments?
Author Response
Dear Reviewer
Attached herewith edited manuscript and response's comment ( table) for your kind perusal.
Thank you so much for your kind comments and suggestion and it has improved a lot of this paper
Thank you
regards
juferi

Round 3
Reviewer 4 Report
After revisions 1 and 2, the paper still suffers from insufficiencies, more especially from several methodological approximations, insufficient assessment of your results, and poor English. Most of the answers provided are partial or inconsistent. Most of the issues have not been addressed. I am sorry of that, meaning that you don’t spend sufficient time to revise the paper before submitting it again. It pushes me to a waste of time for explaining again the main issues that must be solved before approval of the paper. So, please, be so kind to really solve the main issues. For that purpose, I grouped them in the following list:
Global comment- key issue: The English must be improved. Even if you have indicated that it is solved, it is unfortunately not the case. A lot of sentences are approximative with wrong wordings or grammar, or are confusing. This issue is still essential to solve. You must be helped by a native English writer.
Global comment – key issue: You have still a problem with your units.
Line 122: Thanks for modifying the salinity unit. But it is still curious. Are you sure it is in ppt? It would mean brackish water and not freshwater as you mention. Thank you to clarify that issue.
Phosphates and Nitrates units: Several of my questions raised that issue. You only indicate a unit in mg/l whereas both phosphates and nitrates can be expressed as molecules or as atoms, and 1mg/l doesn’t correspond to the same concentration expressed in one or the other. It would be great if you could clarify that.
Line 168: I have already indicated this mistake that is still there. I am sorry but you cannot indicate “concentration of approximately 1.0ml”. A “ml” is a volume not a concentration.
Line 374: You must indicate if in your “mg/ml”, your “mg” are in dry weight or in wet weight.
Global methodological comment: I am sorry, but I still wait you explain why you didn’t conduct your study by using the same water quality for both species. Did you have any reason for not using a single global water sample (that could have been the total volume you needed) with a single water quality for starting the study of the two microalgal species? It would have been much stronger.
Global assessment of the results: There are still some improvements to perform. For instance, I requested a more detailed assessment of your results. Practically you have only indicated a decrease of the concentration with time (lines 354-355) that is a simplified approach. A more in-depth evaluation of the trends could have been performed. However, as you don’t want to include it, let’s as it is. But you should at least solve the two issues:
Line 332-381: Here again it is still one of my preceding comments that has not been solved. How could you have 100% removal of phosphate whereas the final concentration is still higher than 0? 100% removal should mean a “0mg/l” concentration at the end of the test. Your figure 5 is then inconsistent.
Global comment: I have requested the indication on the % of oil that can be extracted from the two considered species (i e their potential for production of bioenergy). I could be wrong, but I didn’t find this information in your paper. The simple indication of nitrates and phosphates decrease is something different. Could you add a sentence on that issue.
One last minor comment: Line 120 “the river water could be contaminated with fresh water originated from upstream (reserve forest hill)…” Logically, a forest has a protecting role on river water quality. Refer to (LAFFORGUE M., 2016. Impact of forests on water quality. In Forest and the Water Cycle: Quantity, Quality, Management, edited by P Lachassagne and M Lafforgue, published by Cambridge Scholars Publishing, pp 204-265).
Author Response
Dear Sir
attached is the response of your comments for your consideration
regards
juferi

Round 4
Reviewer 4 Report
After revisions 1 to 3, the paper has been improved (thanks for that), but it still suffers from insufficiencies. You'll find hereafter my current comments:
Global comment: The English should be improved. It is understandable but some sentences cannot be considered as perfect English.
Global comment – key issue: You have still a problem with your units.
Line 44 and 207-208: regarding the salinity unit, I didn’t request the modification of the salinity unit, but that you verify your results. Indeed, your values mean that you have 90g/l that is more salty than seawater!!
Phosphates and Nitrates units: It is not clear. My comments regard the units, not the mention “ions” that doesn’t solve the issue. I try to be more explicit. Do you have phosphate in mg P/l or in mg PO4/l. Do you have nitrates in mg N/l or in mg NO3/l. Be so kind to precise that in the figures.
Line 377-378: You must indicate “mg dry weight/ml”.
Global methodological comment on the different water quality for the two species cultures: I understand your explanation. My comment was that you should have proceeded differently for a stronger methodology. But I agree that it is presently impossible to modify.
Author Response
Dear Reviewer
Attached herewith table of response with regards your comments and suggestions
Thank you
regards
juferi
